# Nonlinear fusion is optimal for a wide class of multisensory tasks

**Marcus Ghosh[1,2]\*, Gabriel Béna[2], Volker Bormuth[1‡], Dan F. M. Goodman[2‡]**

**1** Laboratoire Jean Perrin, Institut de Biologie Paris-Seine, CNRS, Sorbonne Université, Paris, France,
**2** Department of Electrical and Electronic Engineering, Imperial College London, London, United Kingdom

‡ These authors are joint last authors on this work.
\* mghosh@imperial.ac.uk

**Data Availability Statement:** All code is available at github.com/ghoshm/Nonlinear_fusion. Data are available at doi.org/10.5281/zenodo.12191565.

**Funding:** MG is a Fellow of Paris Region Fellowship Program - supported by the Paris Region, and

## Abstract

Animals continuously detect information via multiple sensory channels, like vision and hearing, and integrate these signals to realise faster and more accurate decisions; a fundamental neural computation known as multisensory integration. A widespread view of this process is that multimodal neurons linearly fuse information across sensory channels. However, does linear fusion generalise beyond the classical tasks used to explore multisensory integration? Here, we develop novel multisensory tasks, which focus on the underlying statistical relationships between channels, and deploy models at three levels of abstraction: from probabilistic ideal observers to artificial and spiking neural networks. Using these models, we demonstrate that when the information provided by different channels is not independent, linear fusion performs sub-optimally and even fails in extreme cases. This leads us to propose a simple nonlinear algorithm for multisensory integration which is compatible with our current knowledge of multimodal circuits, excels in naturalistic settings and is optimal for a wide class of multisensory tasks. Thus, our work emphasises the role of nonlinear fusion in multisensory integration, and provides testable hypotheses for the field to explore at multiple levels: from single neurons to behaviour.

## Author summary

Rather than relying on one sensory modality at a time, animals merge information across their senses and make decisions based on these combined signals. Imagine a predator watching a patch of long grass for prey. The grass moves, indicating the presence of prey, another animal or just the wind. The predator could resolve this ambiguity by combining their visual data, with any coincident sounds or the feel of the wind on their skin. However, how should these signals be combined? Prior work suggests that a sum should be best (i.e. sight + sound). However, we show that this strategy would perform poorly and even fail completely in many multisensory scenarios. Instead, we propose a simple nonlinear function $f$(sight, sound) which excels at these tasks, and could plausibly be implemented by networks of neurons.

funding from the European Union's Horizon 2020 research and innovation program under the Marie Skłodowska-Curie grant agreement No 945298-ParisRegionFP. VB has received funding from the European Research Council (ERC) under the European Union's Horizon 2020 research innovation program, grant agreement number 715980. The funders had no role in study design, data collection or analysis, decision to publish, or preparation of the manuscript.

**Competing interests:** The authors have declared that no competing interests exist.

## Introduction

We continuously detect sensory data, like sights and sounds, and use this information to guide our behaviour [1]. However, rather than relying on single sensory channels, which are noisy and can be ambiguous alone, we merge information across our senses and leverage this combined signal [2]. In biological networks, this process (multisensory integration) is implemented by multimodal neurons, which receive inputs from multiple sensory channels and project to downstream areas. However, how to best describe these network's input-output transformations, which we will term algorithms [3, 4], remains an open question.

As such, a range of algorithms have been proposed [5]. At one extreme, are 1-look algorithms which use only a single channel at a time. Imagine a multimodal network which, due to energy constraints, uses only a single pathway at a time. In the middle are 2-look algorithms which produce an independent output per channel, then use a decision rule to decide between these outputs. For example, in parallel race models, the fastest channel's output is interpreted as the "multimodal" output [6, 7]. At the other extreme are algorithms which fuse information across sensory channels, either linearly [8–11] or non-linearly [5]. In deeper networks this view raises the interesting question of *when* in the sensory hierarchy information should be merged across channels [12, 13]. Though, if all the operations in a network are linear then the order does not matter, and early and late fusion would be equivalent (Section A in S1 File). Consequently, we group these algorithms under the term **linear fusion**.

In theory, nonlinear fusion should outperform linear fusion [5]. However, in practice, both the behaviour of observers and the activity of multimodal neurons seem to approximate linear fusion across a range of classical multisensory tasks. In these tasks, observers are presented with directional cues in two channels and are trained to report the direction with the most evidence. For example, to estimate heading direction from visual and vestibular cues [8–10] or to report the direction of a stimulus from auditory and visual cues [11]. Moreover, in these tasks linear fusion is optimal, in the sense that it generates unbiased (i.e., correct on average) estimates with minimum variance [2].

However, there are two open questions regarding linear fusion: does it generalise beyond the classical tasks used to explore multimodal integration and, what role do neurons with nonlinear responses to multimodal stimuli play? For example, "super-additive" neurons which respond to multimodal stimuli more than predicted by the sum of their unimodal responses [14, 15].

Here, we explore these questions in three steps. First, we begin by considering the role of multimodal neurons. To do so, we develop novel multimodal tasks which focus on the underlying *statistical relationships between channels*, and compare the performance of a range of a models on these tasks. Next, we mathematically derive the optimal algorithm for solving these tasks. Then, finally, in the vein of Marr's *hardware* level [3, 4] we explore how networks can implement this algorithm. Our results lead us to propose a **nonlinear fusion** algorithm for multimodal processing, which outperforms linear fusion and is optimal for a wide-class of multisensory tasks.

## Results

### Multimodal units are not necessary for accuracy or balancing speed/accuracy in classical tasks

We began by training spiking neural networks, via surrogate gradient descent [16], to perform classical multisensory tasks. In these tasks we present sequences of directional information (left / right) in two channels, and train networks to report the direction of overall

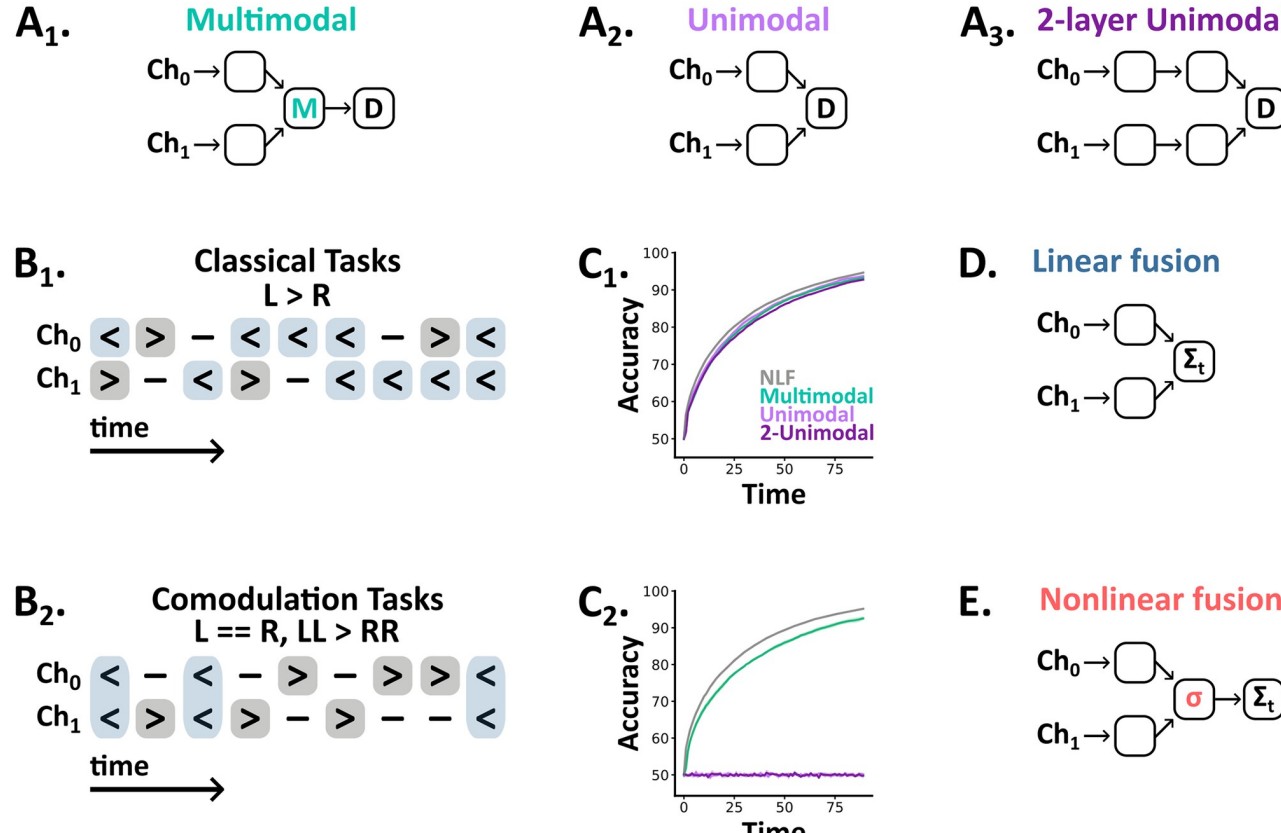

**Fig 1.** **A₁₋₃** Network architectures—spikes flow forward from input channels (Ch₀, Ch₁) to decision outputs (D). **B₁₋₂** Tasks are sequences of discrete symbols (left, neutral, right) per channel, with a fixed number of time steps, and a label (left or right). In classical tasks (B₁) this label indicates the overall bias (e.g. L >R). In comodulation tasks (B₂) this label is encoded jointly across channels (e.g. LL >RR). **C₁₋₂** Test accuracy as a function of time (the number of elapsed time steps within a trial) for the reduced classical (C₁) and probabilistic comodulation (C₂) tasks. We plot the mean (line) and standard deviation (shaded surround) across 5 networks per architecture, plus the optimal accuracy from the nonlinear fusion algorithm (NLF, grey line). **D** In classical tasks the optimal algorithm is to linearly fuse evidence across channels and time. **E** In comodulation tasks nonlinear fusion is optimal—denoted here by σ.

motion (Fig 1B₁). In a reduced case both channels always signal the same direction, while in an extended version we include both unimodal and conflicting trials. Each network was composed of two unimodal areas with feedforward connections to a multimodal layer, connected to two linear output units representing left and right choices (Fig 1A₁). Following training, multimodal networks achieved a mean test accuracy of 93.4% (±0.1% std) on the reduced task and 94.2% (±0.05% std) on the extended task. In line with experimental work [11] their accuracy on the extended task varied by trial type (Fig A in S1 File).

To understand the role multimodal units play in these tasks, we trained two additional sets of networks: one without a multisensory layer (Fig 1A₂) and, to control for network depth, one in which we substituted the multisensory layer for two additional unimodal areas (Fig 1A₃). Surprisingly, these architectures performed similarly to the multimodal architecture on both the reduced (unimodal: 93.9%, ±0.2% std; 2-unimodal: 93.0%, ±0.2% std) and extended task (unimodal: 94.1%, ±0.1% std; 2-unimodal: 94.1%, ±0.1% std), suggesting that multimodal spiking units are not necessary for accuracy on these tasks. So, what computational role do these units play? One alternative is that they are beneficial in balancing speed/accuracy [9]. However, we found that all three architectures accumulated evidence at equivalent rates (Fig 1C₁).

While these results may seem surprising, the optimal strategy in this task simply amounts to linearly fusing evidence across channels and time to form a final estimate. In our models, this **linear fusion** algorithm (Fig 1D) can be directly implemented by the output units, as it is a linear computation, and consequently, the nonlinearity granted by an intermediate multisensory layer provides no benefit. Beyond our models, these results raise the question of when or why biological networks would require multisensory neurons between their unimodal inputs and downstream outputs.

## Multimodal units are critical for extracting comodulated information

To explore when networks need multisensory neurons, we designed a novel task in which we comodulate signals from two channels to generate trials with two properties (Fig 1B$_2$):

1. Within each channel there are an equal number of left and right observations, so a single channel considered in isolation carries no information about the correct global direction.

2. On a fraction of randomly selected time steps (which we term the joint signal strength), both channels are set to indicate the correct direction. Then the remaining observations are shuffled randomly between the remaining time steps (respecting property 1). Thus each trial's label is encoded jointly across channels.

As above, we trained and tested all three architectures on this task. Strikingly, both unimodal architectures remained at chance accuracy (unimodal: 50.4%, ±0.4% std; 2-unimodal: 50.4%, ±0.3% std), while multimodal networks learned the task well (multimodal: 96.0%, ±1.3% std). Additionally, multimodal network accuracy increased in line with joint signal strength (Fig B in S1 File). However, this task seems unrealistic—as it requires a perfect balance between labels—so we developed a probabilistic version with the same constraint (i.e., information is balanced on average but may vary on any individual trial). Again, both unimodal architectures remained at chance accuracy, while multimodal networks approached ideal performance (Fig 1C$_2$).

Together, these results demonstrate that multimodal units are critical for comodulation tasks, though why is this the case? In contrast to classical multisensory tasks, our comodulation tasks require observers to detect coincidences between channels and to assign more evidential weight to these than non-coincident time points. As such, observers must fuse information across channels nonlinearly; an algorithm which we term **nonlinear fusion** (Fig 1E). In our unimodal architectures, fusion happens only at the decision outputs which are linear, meaning they are unable to assign coincidence a higher weight than the sum of the individual observations from each channel. Consequently, the algorithm they implement is equivalent to linear fusion. In contrast, our multimodal, leaky integrate-and-fire, units can assign variable weight to coincidence via their nonlinear input-output function.

To illustrate this difference, consider the following two situations. In situation 1, at time $t = 0$, one channel gives evidence in favour of left, and the other gives no evidence, while at time $t = 1$ the first channel gives no evidence and the other gives evidence of left. In situation 2 at $t = 0$ both channels give evidence of left, and neither channel gives evidence at $t = 1$. To a linear algorithm, these situations provide exactly the same amount of evidence in favour of left as each modality and each time step are independent (and equally reliable by assumption). In contrast, a nonlinear algorithm can assign more weight to the multimodal observation at $t = 0$ in situation 2 than the sum of the unimodal observations in situation 1, and so detect coincidence.

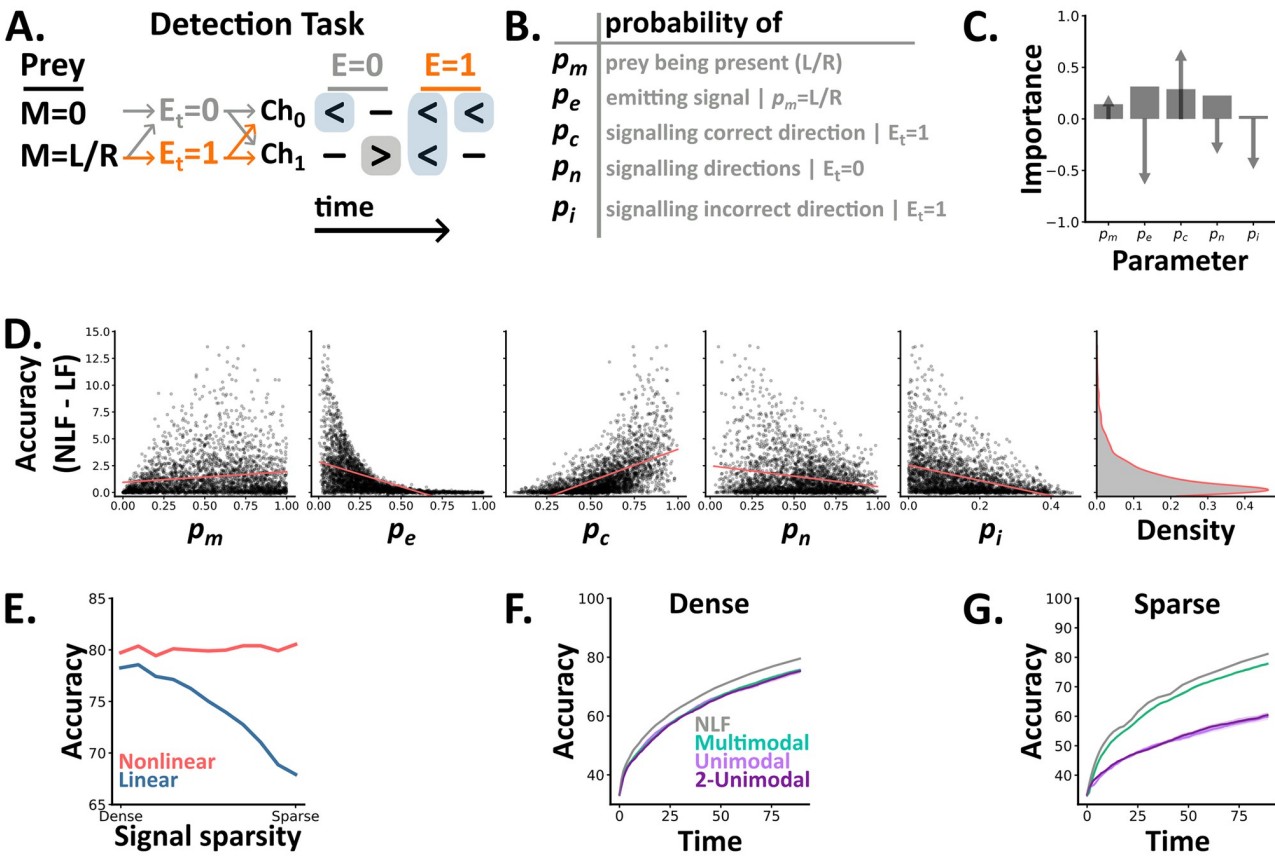

**Fig 2. A-B** In the detection task, a 5 parameter probabilistic model generates trials (sequences of discrete symbols) with different statistics. At each time step $t$ there is either a signal emitted ($E_t = 1$ with probability $p_e$ if there is a prey present) or not ($E_t = 0$) with different probability distributions depending on the value of $E_t$. **C** Each parameter's importance (bars) in predicting the difference in accuracy between the ideal nonlinear and linear models, as determined by a random forest regression model, and each parameters correlation with the difference (arrows). **D** The difference in accuracy between the two algorithms (nonlinear minus linear fusion) as a function of the tasks 5 parameters. For each detection setting (a set of 5 parameters values), we plot the value of each parameter (subplots) and the difference between the two algorithms across 10,000 trials. Overlaid (coral lines) are linear regressors per parameter. The rightmost subplot shows a kernel density estimate of the difference across detection settings. **E** The accuracy of each ideal algorithm across a subset of detection settings where we vary the signal sparsity—a function of $p_e$ and $p_c$. **F-G** The test accuracy over time (the number of elapsed time steps within a trial) of our three SNN architectures, plus the optimal accuracy from the nonlinear fusion algorithm (NLF, grey line), on the two extremes of the family in E.

### Nonlinear fusion excels in naturalistic settings

Our results demonstrate that nonlinear fusion can solve comodulation tasks where linear fusion remains at chance level, but would these two algorithms differ in naturalistic settings? To explore this, we adapted the tasks above to produce a novel detection task in which a predator must use signals from two channels, e.g., vision and hearing, to determine both whether prey are present and if so their direction of motion (3-way classification). In this task, trials are generated via a probabilistic model with 5 parameters which specify the probability of: prey being present in a given trial ($p_m$), emitting cues at a given time when present ($p_e$), signalling their correct ($p_c$) or incorrect ($p_i$) direction of motion and the level of background cues ($p_n$) (Fig 2A and 2B). This task thus closely resembles those above in structure, but with the added realism that information arrives at sparse, unspecified intervals through time.

Using this task, we first compared how the accuracy of the two algorithms differed in distinct settings. To do so, we randomly sampled 10,000 combinations of the task's 5 parameters

and used ideal Bayesian models to determine each algorithm's accuracy. To meaningfully compare the two algorithms we then filtered these results by discarding parameter sets which were either trivially easy or overly difficult (Methods: Accuracy filter). This process left us with 2,836 (28%) sets of parameter combinations. By definition, nonlinear fusion always performs at least as well as linear fusion, and across these settings the median difference (nonlinear minus linear) was 0.73%. Though, the maximum difference was 13.7%, showing that it excels in specific settings.

To understand when nonlinear fusion excels compared to linear fusion, we calculated each parameter's correlation with the difference in accuracy and it's importance in predicting it using a random forest regression model (Methods: Random forest regression) (Fig 2C and 2D). This approach revealed that nonlinear fusion excels when prey signal their direction of motion more reliably (increasing $p_c$), but more sparsely in time (decreasing $p_e$). Critically, these settings resemble naturalistic conditions in which prey provide reliable cues, but try to minimise their availability to predators by remaining as concealed as possible.

Next, we focused on a subset of detection settings by fixing $p_m$, $p_n$ and $p_i$ and identifying combinations of $p_e$ and $p_c$ where nonlinear fusion consistently achieves 80% accuracy. Across this subset, linear fusion's accuracy decreases with the sparsity of the useful signal buried in the noise, varying with both $p_e$ and $p_c$ (Fig 2E). Finally, we trained spiking neural networks on the two extremes of this subset: one in which prey emit unreliable signals, densely in time (**dense detection**) and one in which prey emit reliable signals, sparsely in time (**sparse detection**). In the dense setting, we found no difference in either accuracy or reaction time between the three architectures (Fig 2F). In contrast, the multimodal architecture outperformed both unimodal architectures in the sparse setting (Fig 2G).

Together these results demonstrate that the **nonlinear fusion** algorithm is always as good as **linear fusion**, but excels in naturalistic settings when prey emit reliable signals, sparsely in time. Notably, in natural settings signals from separate channels will likely arrive and be processed at different speeds. However, it is reasonable to assume that these temporal mismatches can be compensated for by appropriate delays.

## The softplus nonlinearity solves a wide class of multimodal tasks

Above, we have focused on tasks in which observers must use information from two sensory channels to reach one of two or three possible decisions. However, even the simplest of organisms are granted more senses and possible behaviours. Thus, we now compare linear and nonlinear fusion in more generalised settings. To do so, we consider the more general case of $N_C$ channels and $N_D$ directions (or classes, more generally).

In the most general case, solving this task would require learning the joint distribution of all variables, i.e. $N_D^{N_C}$ parameters, which would quickly become infeasible as $N_D$ and $N_C$ increase. However, when channels are independent given a shared (underlying time dependent) variable, as in our detection task, it turns out that the optimal solution requires only a small increase in the number of parameters (a fraction $\sim 1/N_D$ more parameters) and adding a single nonlinearity (the softplus function) to the classical linear model (Fig 3A and details in (Methods: Multichannel, multiclass detection task) and (Section E in S1 File)).

So, where does this softplus nonlinearity come from? For the isotropic case (for simplicity), following our derivation (Section E in S1 File) we arrive at the equation:

$$\log P(\text{observation } t \mid M = m) = \log\left(1 + b \exp(c \cdot x(t, m))\right) + \text{ a constant} \qquad (1)$$

where $M$ is the direction being estimated, and $x(t, m)$ is the number of the $N_C$ channels at time $t$ that indicate direction $m$ (that can take any value between 1 and $N_D$). Our observer then

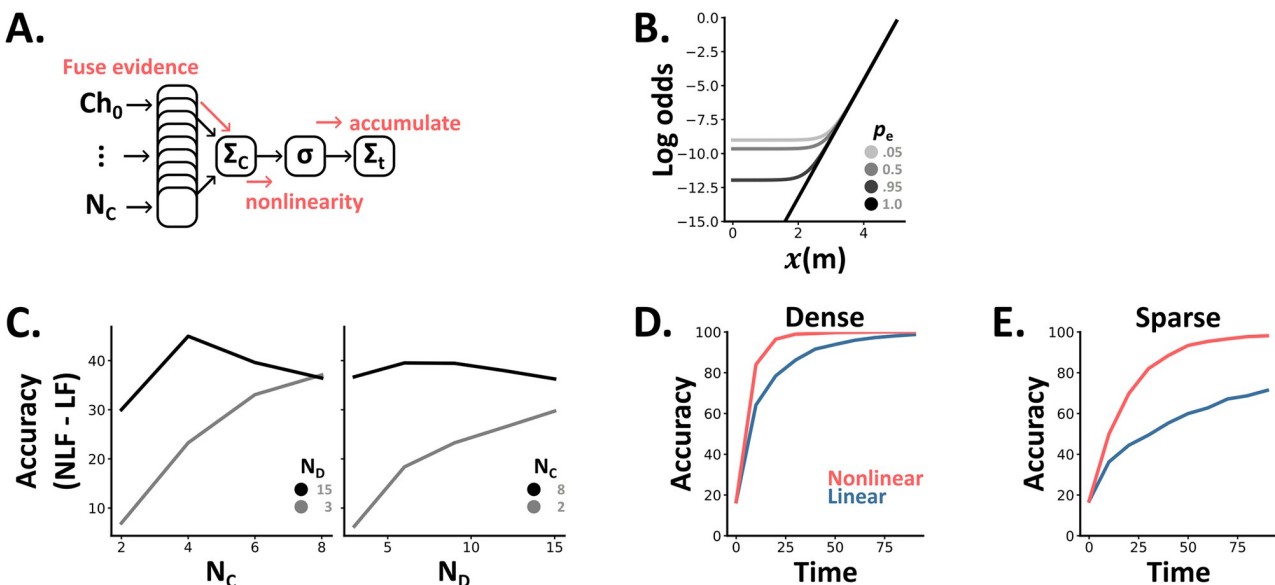

**Fig 3.** **A** In the case of $N_C$(channels) and $N_D$(directions) the optimal algorithm is to sum the evidence for each direction across channels, apply a nonlinearity ($\sigma$) and then accumulate these values. **B** The optimal nonlinearity depends on both the number of channels which indicate the same direction ($x(m)$) and the sparsity of the signal ($p_e$). **C** The difference in accuracy between linear and nonlinear fusion (LF and NLF) depends on both the number of channels and directions. **D-E** Accuracy over time (the number of elapsed time steps within a trial) curves for linear (blue) and nonlinear (coral) fusion in dense (D) and sparse (E) settings with 5 channels and 6 directions.

estimates $M$ by computing $x(t, m)$ at each time $t$ via a linear summation across channels, passing it through the nonlinear softplus function (softplus($x$) = log(1 + $b$ exp($c \cdot x$))), summing these values linearly over time to get the log odds of each possible direction $m$ and then estimating $M$ to be the value of $m$ which gives the maximum value (Fig 3A). This provides us with two insights:

1. As the number of channels ($N_C$) increases, the function gets closer to a shifted rectified linear unit (ReLU). That is, it gets close to returning the value 0 when $x$ is less than some threshold ($x_{\text{thresh}}$), and afterwards is proportional to $x - x_{\text{thresh}}$. Consequently, with many channels, the optimal algorithm should ignore time windows ($t$) where the evidence is below a certain threshold $x_{\text{thresh}}$, and linearly weight those above (Fig 3B).

2. As the input becomes more dense, in our case as $p_e$ approaches 1, the optimal algorithm becomes entirely linear. As such, we can see why classical multisensory studies, which have focused on dense tasks, concluded that the linear fusion was optimal (Fig 3B).

Returning to our ideal observer models, in these extended settings we found that the difference between the two algorithms (nonlinear minus linear) increases with both the number of directions ($N_D$) and channels ($N_C$) (Fig 3C). As above (Results: Nonlinear fusion excels in naturalistic settings), we observe little difference in dense settings (Fig 3D) but large differences, up to 30% improvements of nonlinear over linear, in sparse settings (Fig 3E).

Finally, we adapted the observations in this multi-direction, multi-channel task, from discrete to continuous values with any probability distribution (Methods: Continuous detection task) and show results for the Gaussian case. Mathematically, the optimal algorithm in the Gaussian case has the form softplus(quadratic(observations, $m$)). While this demonstrates that the exact form of the optimal function will depend upon the distribution of each channel's

signals; it suggests the softplus function will suit a wide-class of multisensory tasks. Extending our ideal observer models to this continuous case, generated similar results to the discrete case. That is, the difference between the two algorithms was negligible in dense, but large in sparse, settings (Fig C in S1 File).

Together, these results explain why prior studies have focused on linear fusion, and demonstrate how just a small increase in algorithmic complexity (from linear to nonlinear fusion) leads to large improvements in performance across a wide-class of multisensory tasks.

## Network implementations of nonlinear fusion

Above, we demonstrated that the softplus nonlinearity is the optimal algorithm for a wide-class of multisensory tasks. Here, in the vein of Marr's *hardware* level [3, 4], we explore how networks can implement this algorithm and how one can distinguish which algorithm (linear or nonlinear fusion) an observer's behaviour more closely resembles.

**Network behaviour is robust across nonlinearities.** To explore how precisely networks need to approximate the softplus function (or if any nonlinearity will do) we trained minimal artificial neural networks on our two channel tasks, and compared networks whose multi-modal units used linear, rectified linear, sigmoidal or softplus activation functions (Methods: Artificial neural networks). In tasks with little joint signal across across channels (classical and dense-detection), linear activations were sufficient (Fig 4A). In contrast, tasks with more joint signals (sparse-detection and comodulation) required non-linear activations, though all three

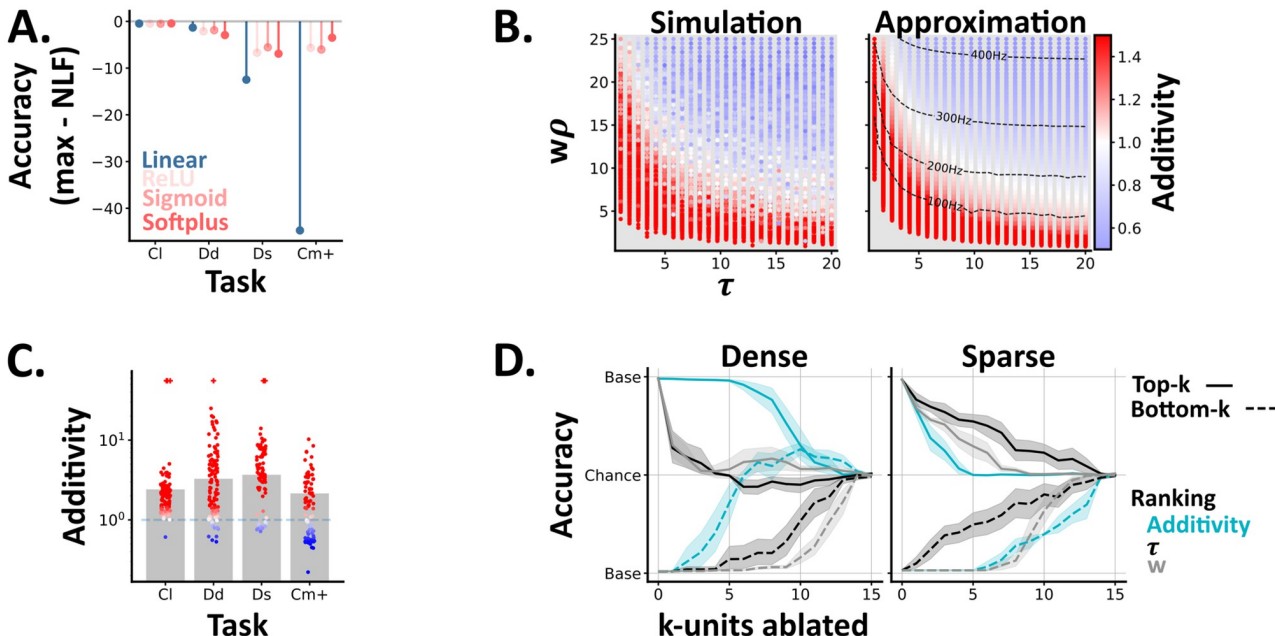

**Fig 4. A** ANN results. For each task (Cl: reduced classical, Dd: dense detection, Ds: sparse detection, Cm+: probabilistic comodulation) and activation function (colours) we plot the maximum test accuracy (across 5 networks) minus the optimal nonlinear fusion accuracy (NLF). **B** Single unit model results. Each point shows the models additivity as a function of the membrane time constant ($\tau$) and mean input ($w\rho$). The grey underlay shows when the multisensory unit fails to spike. The left panel shows the average additivity from 10 simulations. The right shows the results from an approximation method, with the multisensory unit's firing rate (Hz) overlaid. **C-D** SNN results. **C** The additivity of each multisensory unit (circles) from spiking networks trained on different tasks. Bars indicate the mean additivity (across networks) per task. Colours, per unit, are as in B. Units which spike to multi-, but neither unisensory stimulus are denoted with a plus symbol. **D** Network accuracy, from baseline to chance, as we ablate either the top (solid lines) or bottom (dashed lines) k-units, ranked by different unit properties (colours). We plot the mean and std across networks, and invert the y-axis for the bottom-k results. Left / right—networks trained and tested on dense or sparse detection.

non-linearities were equivalent in terms of both accuracy and reaction speeds (Fig D in S1 File); suggesting that any non-linear function may be sufficient. Though, how do these mathematical functions relate to the activity patterns of real spiking neurons?

**Simple, single neuron models generate sub- to super-additive states.** In experimental studies [14] the input-output functions of individual multimodal neurons are often inferred by comparing their multimodal response $f(Ch_0, Ch_1)$ to the sum of their unimodal responses $f(Ch_0, 0) + f(0, Ch_1)$ via a metric known as **additivity**:

$$\text{additivity} = \frac{f(Ch_0, Ch_1)}{f(Ch_0, 0) + f(0, Ch_1)} \tag{2}$$

Using this metric, neurons can be characterised as being in sub- or super-additive states (i.e., outputting less or more spikes than expected based on their unimodal responses) [14]. However, the link between these **neuron states** and **network behaviour** remains unclear [5]. To explore this we conducted two experiments; one at a single neuron level (here) and one at the network level (Results: Unit ablations demonstrate a causal link between additivity and network behaviour).

To understand how these states arise in spiking neurons, we simulated single multimodal units (Methods: Simulation), with differing membrane time constants ($\tau$) and mean input weights ($w$), and calculated their additivity as we varied the mean firing rates of their input units ($\rho$) (Fig 4B). These simulations recapitulated two experimental results, and yielded two novel insights. In agreement with experimental data [14] most units (60%) exhibited multiple states, and we found that lower input levels (both weights and firing rates) led to higher additivity values (Fig 4B); a phenomenon termed *inverse effectiveness* [17]. Moving beyond experimental data, our simple, feedforward model generated all states, from sub- to super-additive; suggesting that other proposed mechanisms, such as divisive normalisation [18], may be unnecessary in this context. Further, we found that units with shorter membrane time constants, i.e. faster decay, were associated with higher additivity values (Fig 4B); suggesting that this may be an interesting feature to characterise in real multimodal circuits. Notably, an alternative modelling approach (Methods: Diffusion approximation), in which we approximated the firing rates of single multimodal neurons [19, 20], generated almost identical results (Fig 4B-right).

**Unit ablations demonstrate a causal link between additivity and network behaviour.** To understand how these unit states relate to network behaviour we calculated the additivity of the multimodal units in our trained spiking neural networks, and compared these values across tasks. We found that most units were super-additive, though observed slight differences across tasks (Fig 4C); suggesting a potential link between single unit additivity and network behaviour.

To test this link, we ranked units by their additivity (within each network), ablated the highest or lowest $k$ units and measured the resulting change in test accuracy. On the dense detection task, we found that ablating the units with lowest additivity had the greatest impact on performance, while on sparse detection we observed the opposite relation (Fig 4D). The classical and probabilistic comodulation tasks respectively resembled the dense and sparse detection cases (Fig E in S1 File). To understand these relations further we then ranked unit's by their membrane time constants ($\tau$) or mean input weights ($w$) and repeated the $k$-ablations. On both tasks ablating units with high input weights and / or long $\tau$ significantly impaired performance; highlighting the importance of *accumulator-like* units for both tasks. However, on sparse detection, we also observed that ablating units with short $\tau$ had a symmetrical effect (Fig 4D); suggesting an additional role for *coincidence-detector-like* units on this task.

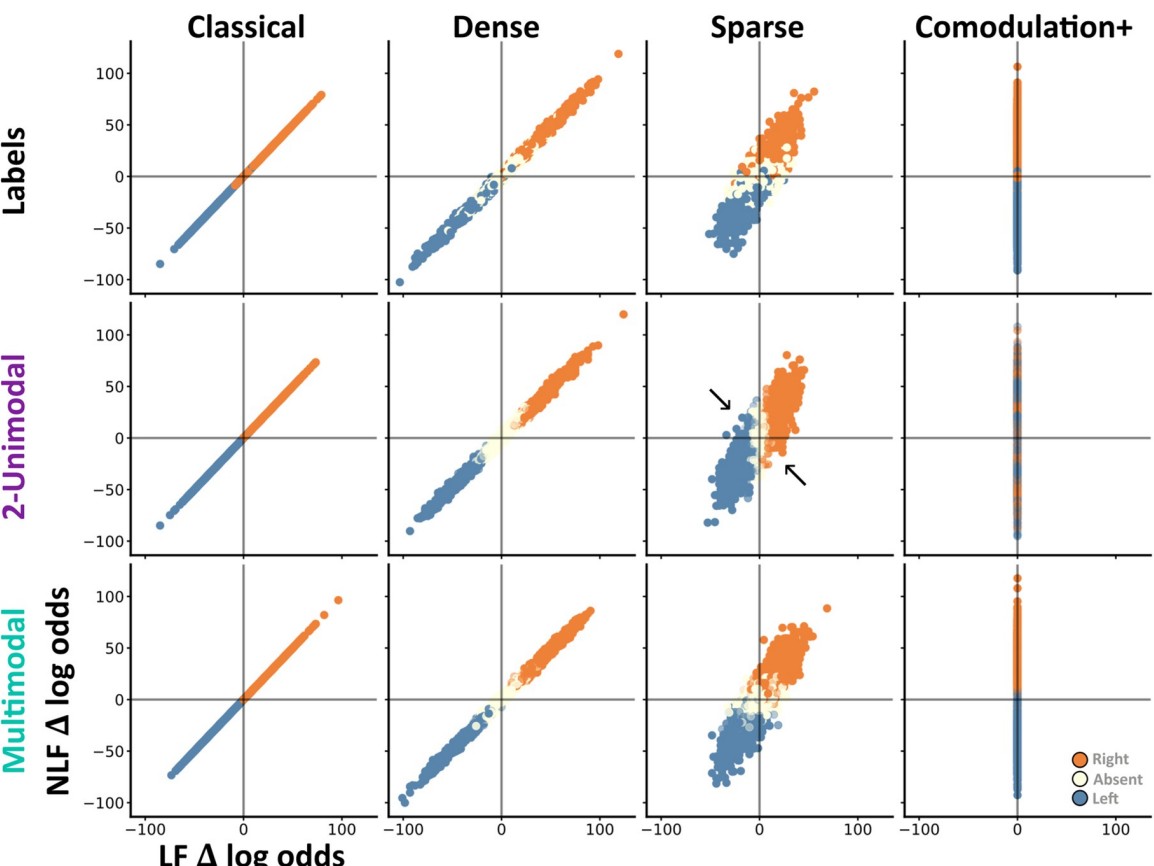

**Fig 5.** For each subplot we scatter 2000 random trials on the same two axis: the amount of right minus left evidence according to each algorithm (Δ log odds). Thus, when x = 0 the trial contains an equal amount of left and right information according to the linear fusion algorithm. Each column contains trials from a different task (reduced classical, dense detection, sparse detection, probabilistic comodulation). The top row shows each trial's label (left = blue, absent = off-white, right = orange). The middle and last row show the most common choice across 10 SNNs of either the 2-layered unimodal (middle) or multimodal (bottom) architecture. For these rows each trial's alpha indicates how consistently networks chose the most common value. For example, on probabilistic comodulation the 2-unimodal networks choose randomly—and so these trials have a low alpha. Otherwise networks choose consistently—so most trials have high alpha. Black arrows highlight subsets of trials where the two architectures choose opposite directions.

From these results, we draw two conclusions. First, different multimodal tasks require units with different properties (e.g. long vs short membrane time constants). Second, additivity can be used to identify the most important units in a network. However, as a unit's additivity is a function of its intrinsic parameters ($\tau$, $w$, $\rho$), additivity is best considered a proxy for these informative but harder to measure properties. Notably, an alternative approach in which we used combinatorial ablations (i.e. ablating unit 1, 1–2, 1–3 etc) to calculate each unit's causal contribution to behaviour [21], yielded similar results (Fig F in S1 File).

**Behaviour can distinguish which algorithm an observer is implementing.**   Finally, when testing an observer on a task, we wish to distinguish if their behaviour more closely resembles either algorithm (linear or nonlinear fusion). To do so, we measured the amount of evidence which each algorithm assigns to each direction (difference in log odds of direction left or right, given the observations), per trial, and then scattered these in a 2d space.

From this approach we garner two insights. First, by colouring each trial according to it's ground truth label, we can visualise both the information available on each trial and the relations between tasks (Fig 5). For example, in the classical task both algorithms assign the same

amount of evidence to each direction, so each trial lies along $y = x$. In contrast, in the probabilistic comodulation task only nonlinear fusion is able to extract any information, and all trials lie along the y-axis. Second, by colouring each trial according to an observers choices, we can compare how closely their behaviour resembles either algorithm (Fig 5). Applying this approach to our 2-layer unimodal, and multimodal SNN architectures illustrates that while their behaviour is indistinguishable on the classical and dense-detection tasks, there are a subset of sparse detection trials where the two architectures choose opposite directions as their behaviour is respectively closer to the linear or nonlinear fusion algorithms (Fig 5). These trials are essentially linear-nonlinear conflict trials as there is an overall bias in one direction (e.g. L > R) but more coincident information in the other (e.g. LL < RR).

Thus, coupled with task accuracy—which is all that is necessary on our comodulation tasks—trial-by-trial choices are sufficient to distinguish whether an observer's behaviour more closely resembles linear or nonlinear fusion.

## Discussion

Prior experimental and theoretical work suggests that multimodal neurons linearly fuse unimodal information across channels. In contrast, our results, from three levels of abstraction, suggest that fusion may be nonlinear. Resolving *which algorithm better describes multimodal processing in biological systems* will require further experimental work. Nevertheless, here, we argue that nonlinear fusion is likely to be a better description.

On classical tasks, both algorithms are indistinguishable and so describe multimodal processing equivalently. As such, prior experimental results are consistent with either algorithm. In contrast, on our novel comodulation tasks, linear fusion remains at chance level, while nonlinear fusion is optimal. Thus, these tasks constitute simple experiments to determine which algorithm better describes an observer's behaviour. However, both the classical and comodulation tasks seem unrealistic for two reasons. First, both are composed of dense signals (i.e. every time step is informative). Second, both treat the relations between channels in extreme ways: in the former, the temporal structure of the joint multimodal signal carries no information, while in the latter all the information is carried in the joint signal and the task is impossible using only one modality.

In contrast, in our detection-based tasks observers must extract periods of signal from background noise, and both the observations *within* and *across* channels are informative. As such, these tasks seems more plausible, particularly given the added realism that there is sometimes nothing to detect, and the fact that both algorithms can solve them to some extent. Moreover, unlike pure multisensory synchrony or coincidence detection tasks [22] in which the observer must *explicitly* detect coincidence, here coincidence is an informative, *implicit* feature. Nonlinear fusion is a generalisation of linear fusion and so always performs at least as well on our detection tasks. Though, nonlinear fusion excels in realistic cases where prey signal their direction of motion reliably but sparsely in time, and in cases where the number of directions or channels is high. Note that while channels could originate from separate modalities, like vision or sound, our analysis also extends to independent sources of information from within modalities; so the number of channels may exceed the number of modalities. In sum, our results suggest that nonlinear fusion could provide significant benefits in naturalistic settings and, assuming that performance is ecologically relevant, may more plausibly describe multisensory behaviour.

Though, there are four limitations to consider here. First, we focused on discrete stimulus tasks with a fixed trial length. This is un-naturalistic as tasks in the real-world are likely to be temporally unbounded and composed of continuous stimuli. However, our decoding of

observer's accuracy over time hints at their performance in free-reaction tasks, and extending part of our work to the continuous case yielded similar results. In a similar vein, extending our work to consider non-evidence accumulation-based tasks would further generalise our results. Second, we focused on small network models with simple architectures. This is problematic as it elides the complexity of real multimodal circuits, like the presence of skip connections [23], and may make our results difficult to relate to real circuits. However, our multimodal architecture is optimal for all the tasks we considered—suggesting that additional complexity may only be necessary for even more challenging tasks, and the multimodal units in our model simply represent the first neurons to receive inputs from multiple modalities—wherever they may be found in the brain. Third, we only considered equally reliable channels. In the tasks we consider, it would be straightforward to model channels with differing reliability via some additional parameters, though it would not substantially change our results. Moreover, reliability is not well defined for non-independent channels (Section B in S1 File). Finally, we assumed that observations are generated by a single underlying cause. Relaxing this assumption and extending our work to consider causal inference is an exciting future direction [24–26].

Beyond behaviour, are there hardware features which make either algorithm a better description of multimodal processing? In biological systems, these algorithms must be implemented by networks of spiking neurons. As neurons naturally transform their inputs via a spiking nonlinearity, and we show that the exact form of this nonlinearity is not critical, nonlinear fusion constitutes a more natural solution for networks of spiking neurons. Concurrently, the behaviour of trained SNNs more closely resembles nonlinear fusion on both our sparse-detection and comodulation tasks.

Ultimately, our work demonstrates that extending the linear fusion algorithm—with only a few additional parameters and a single nonlinearity—results in an algorithm (nonlinear fusion) which is optimal for a wide class of multimodal tasks; and may constitute a better description of multimodal processing in biological systems.

## Methods

In short, we introduce a family of related tasks (Methods: Tasks). In general, observers must infer a target motion (left or right) from a sequence of discrete observations in two or more channels—which each signal left, neutral or right at each time step. To model channels with equal reliability, we simply generated each channel's data using the same procedure. For each task we computed ideal performance using maximum a posteriori estimation (Methods: Bayesian models). When working with spiking neural networks (Methods: Spiking neural networks) we represented task signals via two populations of Poisson neurons per channel ($Ch_0$) and ($Ch_1$) with time-varying and signal-dependent firing rates. In our spiking networks we modelled hidden units as leaky integrate-and-fire neurons (Methods: Single spiking neurons). Readout units were modelled using the same equation but were non-spiking. To ensure fair comparisons between architectures, we varied the number of units to match the number of trainable parameters as closely as possible. We trained networks using Adam [27], and in the backward pass replaced the derivative with the SuperSpike surrogate function [16]. For all conditions we trained 5–10, networks, and report the mean and standard deviation across networks for each comparison.

### Tasks

For all tasks, there is a target motion to be inferred, represented by a discrete random variable $M$. In addition, there is always a sequence of discrete time windows $t = 1, .., n$. At each time window $t$, observations $C_t^i$ are made in each channel $i = 1, . . ., N_C$. In the case of two channels

we will sometimes refer to $C_t^1 = A_t$ and $C_t^2 = V_t$ (evoking auditory and visual channels). Observations at different time windows will be assumed to be independent (except for the perfectly balanced comodulation task). For simplicity, we usually assume that information from each channel is equally reliable, and that each direction of target motion is equally likely, but these assumptions can be dropped without substantially changing our conclusions. Sample Python code for each task can be found in Section D in S1 File. Our working code can be found at http://github.com/ghoshm/Nonlinear_fusion.

**Classical task.** In the *classical task* we consider two channels, allow $M$ to take values $\{-1, 1\}$ representing left and right with equal probability, and assume that the channels are conditionally independent given $M$. We define a family of tasks via a *signal strength* $0 \leq s \leq 1$ that gives the probability distribution of values of a given channel at a given time to be:

$$\begin{aligned} p_c(s) &= P(C_t^i = M \mid M) &= (1 + 2s)/3 &\quad \text{correct} \\ p_i(s) &= P(C_t^i = -M \mid M) &= (1 - s)/3 &\quad \text{incorrect} \\ p_n(s) &= P(C_t^i = 0 \mid M) &= (1 - s)/3 &\quad \text{neutral} \end{aligned} \tag{3}$$

This has the properties that (a) when signal strength $s = 0$ each value is equally likely ($p_c = p_i = p_n$) and the task is impossible, (b) as signal strength increases the chance of correct information increases and the chance of neutral or incorrect information decreases, (c) when signal strength $s = 1$ all information is correct ($p_c = 1$, $p_i = p_n = 0$). By default we use $s = 0.1$ for this task.

**Extended classical task.** In the *extended classical task* there are motion directions $M_A$, $M_V$ for each channel ($M_{A/V} = \pm 1$ with equal probabilities), along with signal strengths $s_A$, $s_V \in [0, 1]^2$ which both vary across trials. The overall motion direction $M$ is the direction of whichever modality has the higher signal strength in a given trial—we exclude trials when $s_A = s_V$.

$$M = \begin{cases} M_A & \text{with probability 1 if } s_A > s_V \\ M_V & \text{with probability 1 if } s_V > s_A \end{cases} \tag{4}$$

We also includes *uni-sensory* trials, when $s_A = 0$ or $s_V = 0$. The probability distributions are the same as in (3) except with $s = s_A$ for $i = 1$ and $s = s_V$ for $i = 2$.

**Perfectly balanced comodulation task.** In the *perfectly balanced comodulation task* we use two channels and allow $M = \pm 1$ as in the classical task. We guarantee that in each channel, the number of left (-1), neutral (0) and right (+1) observations are precisely equal in number (one third of the total observations in each case). We define a signal strength $0 \leq s \leq 1/3$ and randomly select $sn$ of the $n$ time windows $t$ in which we force $A_t = V_t = M$. We set the remaining values of $A_t$ and $V_t$ randomly in such a way as to attain the per-channel balance in the number of observations of each value. Note that this is the only task in which observations in different time steps are not conditionally independent given $M$.

**Probabilistically balanced comodulation task.** The *probabilistically balanced comodulation task* is designed on the same principle as the perfectly balanced comodulation task but rather than enforcing a strict balance of left and right observations we only enforce an expected balance across trials, and this allows us to reintroduce the requirement that different time steps are independent. We define the notation $p_{av} = P(A_t = k_a M, V_t = k_v M|M)$ where $a, v$ can take values $c$ (correct), $n$ (neutral) and $i$ (incorrect), and $k_c = 1$, $k_i = -1$ and $k_n = 0$. We assume that the two channels are equivalent so $p_{av} = p_{va}$ and (to reduce the complexity) we assume $p_{ci} = p_{ic} = p_{nn} = 0$ since these cases carry no information about $M$. The balance requirement gives us the equation $2p_{cc} + p_{cn} + p_{nc} = 2p_{ii} + p_{in} + p_{ni}$ or (by symmetry) $p_{cc} + p_{cn} = p_{ii} + p_{in}$. This gives us

a two parameter family of tasks defined by these probabilities, and we choose a linear 1D family defined by a signal strength $s$ as follows:

$$
\begin{aligned}
p_{cc} &= (1/3)s + (1/9)(1-s) \\
p_{ii} &= (1/9)(1-s) \\
p_{cn} &= (1 + p_{ii} - 3p_{cc})/4 \\
p_{in} &= (1 + p_{cc} - 3p_{ii})/4
\end{aligned}
\tag{5}
$$

This has the properties that when $s = 0$ the task is impossible ($p_{cc} = p_{ii}$ and $p_{cn} = p_{in}$) and when $s = 1$ the probability $p_{cc}$ takes the highest possible value it can take (1/3). By default, we use $s = 0.2$ for this task.

**Detection task.** In the *detection task* we have two channels and now allow for the possibility $M \in \{-1, 0, 1\}$ where $M = 0$ represents the absence of a target. We introduce an additional variable $E_t \in \{0, 1\}$ which represents whether the target is emitting a signal ($E_t = 1$) or not ($E_t = 0$). If $M = 0$ then $E_t = 0$ for all $t$, and if $M \neq 0$ then $E_t$ is randomly assigned to 0 or 1 at each time step. When $E_t = 0$ the probabilities of observing a left/right in any given channel are equal, whereas when $E_t = 1$ you are more likely to observe a correct than incorrect value in a given channel. Channels are conditionally independent given $M$ and $E_t$ but dependent given only $M$. We can summarise the probabilities as follows:

$$
\begin{aligned}
p_m &= P(M \neq 0) && \text{target present} \\
p_e &= P(E_t = 1 | M \neq 0) && \text{emission} \\
p_n &= P(A_t \neq 0 \mid E_t = 0, M) = P(V_t \neq 0 \mid E_t = 0, M) && \text{noise} \\
p_c &= P(A_t = M \mid E_t = 1, M) = P(V_t = M \mid E_t = 0, M) && \text{signal correct} \\
p_i &= P(A_t = -M \mid E_t = 1, M) = P(V_t = -M \mid E_t = 0, M) && \text{signal incorrect}
\end{aligned}
\tag{6}
$$

If we set the probability of a target being present $p_m = 1$ and the emission probability $p_e = 1$ then this reduces to the classical task.

We create a 1D family of detection tasks by first fixing $p_m = 2/3$ (all values of $M$ equally likely), $p_n = 1/3$ (all observations equally likely when signal not present), $p_i = 0.01$ (signal reliable when present), and then picking from the smooth subset of values of $p_c$ and $p_e$ that give the ideal nonlinear fusion algorithm a performance level of 80%. We set $s = 0$ for the point with the lowest value of $p_c$ and highest value of $p_e$, and $s = 1$ for the opposite extreme.

**Multichannel, multiclass detection task.** We generalise the detection task to $N_D$ directions so $M \in \{1, \ldots, N_D\}$ and $N_C$ channels which can take any of these $N_D$ values, so $C_t^i \in \{1, \ldots, N_D\}$ for $i = 1, \ldots, N_C$. We now assume there is always a target present and set $P(E_t = 1) = p_e$. We make an isotropic assumption that every direction is equally likely (although see Section E in S1 File for the non-isotropic calculations). In this case, when $E_t = 0$ every observation is equally likely. When $E_t = 1$ we let $p_c$ be the probability of a correct observation, and all other

observations are equally likely. In summary:

$$P(C_t^i = j \mid E_t, M) = \begin{cases} p_c & \text{if } E_t = 1 \text{ and } j = M \\ (1 - p_c)/(N_D - 1) & \text{if } E_t = 1 \text{ and } j \neq M \\ 1/N_D & \text{if } E_t = 0 \end{cases} \tag{7}$$

**Continuous detection task.**   In the continuous detection task we allow the same set of values of $M \in \{-1, 0, 1\}$ as in the detection task, and the same definition of $E_t$, but we now allow $N_C$ channels and each channel is a continuous variable that follows some probability distribution. For the general case, see the calculations in Section E in S1 File, but here we make the assumption that channels are normally distributed. If $E_t = 0$ then $C_t^i \sim N(0, 1)$ and if $E_t = 1$ then $C_t^i \sim N(\mu M, \sigma^2)$. By default we assume $p_m = 2/3$ and $\mu = 0.5$, then vary $p_e$ and $\sigma$ for the dense ($p_e = 0.5$, $\sigma = 1.0$) and sparse ($p_e = 0.05$, $\sigma = 0.1$) cases.

## Bayesian models

We define two maximum a posteriori (MAP) estimators for all tasks except the perfectly balanced comodulation task (because they assume that all time windows are conditionally independent given $M$). The estimator is:

$$\hat{M} = \text{argmax}_m P(M = m \mid \mathbf{C}) \tag{8}$$

Here $\mathbf{C}$ is the vector of all observations $C_t^i$. We give complete derivations in Section E in S1 File, but in summary this is equivalent to the following that we call the ideal nonlinear fusion estimator:

$$\hat{M} = \text{argmax}_m \left( \log P(M = m) + \sum_{t=1}^{n} \log P(\mathbf{C}_t \mid M = m) \right) \tag{9}$$

If we additionally assume that channels are conditionally independent given $M$ (which is true for some tasks but not others) we get the classical estimator that we call linear fusion:

$$\hat{M} = \text{argmax}_m \left( \log P(M = m) + \sum_{i=1}^{N_C} \sum_{t=1}^{n} \log P(C_t^i \mid M = m) \right) \tag{10}$$

We call this estimator linear fusion because each modality can accumulate the within-channel evidence $\epsilon^i(m) = \sum_{t=1}^{n} \log P(C_t^i \mid M = m)$ separately before it is linearly summed across channels to get $\epsilon(m) = \log P(M = m) + \sum_{i=1}^{N_C} \epsilon^i(m)$. In the more general case addressed by the nonlinear fusion estimator, information from across channels needs to be nonlinearly combined at each time $t$ to compute $\log P(\mathbf{C}_t|M = m)$ before it is accumulated across time.

Note that when $M \neq 0$ and $M = \pm 1$ are equally likely, this is akin to a classical drift-diffusion model. Let $\delta_t$ be the difference in evidence at time $t$ between right and left, $\delta_t = \log P(\mathbf{C}_t|M = 1) - \log P(\mathbf{C}_t|M = -1)$. The decision variable given all the evidence up to time $s$ is $D_s = \sum_{t=1}^{s} \delta_t$ which jumps in the positive direction when evidence in favour of motion right is received, and in the negative direction when evidence in favour of motion left is received. The estimator in this case is $\hat{M} = \text{sign } D(n)$, i.e. right if the decision variables ends up positive, otherwise left.

In the classical task, this estimator simplifies to computing whether or not the number of times left is observed across all channels is greater than the number of times right is observed, and estimating left if so (or right otherwise). In the more general discrete cases you count the

number of times each possible vector of observations across channels $\mathbf{C}_t$ occurs and compute a weighted sum of these counts. In cases where it is not feasible to exactly compute the ideal weights, we can use a linear classifier using these vectors of counts as input, and we use this to approximate the linear and nonlinear fusion estimators for the perfectly balanced comodulation task.

### Artificial neural networks

Each minimal network was composed of: four unimodal units, two multimodal units and three decision outputs (prey-left, prey-absent or prey-right), connected via full, feedforward connections. Unimodal units were binary, and each sensitive to a single feature (e.g. channel 1 —left). Multimodal units transformed their weighted inputs via one of the following activation functions: linear, rectified linear (ReLU), sigmoid, or softplus of the form:

$$y = \log\left(1 + e^{ax+b}\right) \tag{11}$$

where $a$ and $b$ are trainable parameters. To ensure fair comparisons across activations, we added two trainable biases to the other activations, such that all networks had a total of 16 trainable parameters.

To read out a decision per trial, we summed the activity of each readout unit over time, and took the argmax. To train networks, we initialised weights uniformly between 0 and 1, both $a$ and $b$ as 1, and used Adam [27] with: lr = 0.005, betas = (0.9, 0.999), and no weight decay.

### Single spiking neurons

**Simulation.** We modelled each spiking unit as a leaky integrate-and-fire neuron with a resting potential of 0, a single membrane time constant $\tau$, a threshold of 1 and a reset of 0. Simulations used a fixed time step of dt = 1ms and therefore had an effective refractory period of 1ms.

$$
\begin{aligned}
\tau\frac{dv}{dt} &= -v & &\text{continuous time dynamics} \\
v &\leftarrow v + w & &\text{on receiving a spike at synapse with weight } w \\
v &\geq 1 & &\text{condition for generating a spike} \\
v &\leftarrow 0 & &\text{after generating a spike}
\end{aligned}
\tag{12}
$$

To generate results for our single unit models (Results: Simple, single neuron models generate sub- to super-additive states) we simulated individual multimodal units receiving Poisson spike trains from 30 input units over 90 time steps. We systematically varied three parameters in this model: the multimodal unit's membrane time constant ($\tau$: 1–20ms), its mean input weights ($w$: 0–0.5) and the mean unimodal firing rate ($\rho$: 0–10Hz). We present the average results across 10 simulation repeats.

**Diffusion approximation.** As an alternative approach (Results:Simple, single neuron models generate sub- to super-additive states), we approximated the firing rates of single multimodal neurons using a diffusion approximation [19, 20]. In the limit of a large number of

inputs, the equations above can be approximated via a stochastic differential equation:

$$\tau \frac{dv}{dt} = \mu - v + \sigma\sqrt{\tau}\xi$$
$$\mu = \sum w\rho\tau$$
$$\sigma^2 = \sum w^2\rho\tau$$

(13)

where $\xi$ is a stochastic differential that can be thought of over a window $[t, t + \delta t]$ as a Gaussian random variable with mean 0 and variable $1/\sqrt{\delta t}$, and $w$ and $\rho$ are the weights and firing rates of the inputs. Using these equations we calculated the firing rates of single units:

$$\text{ISI} = \tau\sqrt{\pi} \int_{-\mu/\sigma}^{(1-\mu)/\sigma} e^{x^2}(1 + \text{erf}(x))dx$$
$$\text{FR} = 1/(\text{ISI} + t_{\text{refractory}})$$

(14)

We computed this for both multimodal and unimodal inputs with $\rho_{\text{unimodal}} = \rho_{\text{multimodal}}/2$, and calculated additivity as the multimodal firing rate divided by twice the unimodal firing rate.

## Spiking neural networks

**Input spikes.** We converted the tasks' *directions per timestep (A, V)* into spiking data. Input data takes the form of two channels of 196 units, each of them sub-divided again in two equal sub-populations representing left or right. Then, at each timestep $t$, each unit's probability of spiking depends on the underlying direction of the stimulus at time $t$ ($A_t$, $V_t$) and spike rates $p_{\text{min}}$, $p_{\text{max}}$:

$$p_{\text{Ch}}^i = \begin{cases} p_{\text{min}} & \text{if } i \neq \text{Ch}_t \\ p_{\text{max}} & \text{if } i = \text{Ch}_t \end{cases}, \text{where } i \in \{L, R\} \text{ is the subpopulation, and } \text{Ch} \in \{A, V\} \quad (15)$$

From those probabilities at each timesteps, we generate the two populations of spikes, resulting in Poisson-distributed spikes with rates depending on the underlying signal (Fig 6).

**Spiking units.** We modelled each unit as in Methods: Simulation. Both uni- and multi-modal units were initialised with heterogeneous membrane time constants drawn from a gamma distribution centred around $\tau = 5$ms and clipped between 1 and 100ms. Readout units were modelled using the same equation, but were non-spiking and used a single membrane time constant $\tau_r = 20$ms.

**Architectures.** In our **multimodal architecture**, *196* input units sent full feed-forward connections to *30* unimodal units, per channel. In turn, both sets of unimodal units were fully connected to *30* multimodal units. Finally, all multimodal units were fully connected to two

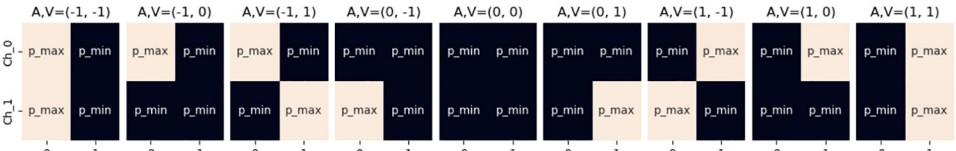

**Fig 6.** Average spike rates for different channel local directions.

readout units representing left and right outputs. Thus, our multimodal architecture had a total of *13620* trainable parameters. To ensure fair comparisons between architectures, we matched the number of trainable parameters, as closely as possible, by varying the number of units in our unimodal architectures. In our **unimodal architecture** we used *35* unimodal units per channel and no multimodal units (*13790* trainable parameters). In our **double-layered unimodal architecture** we replaced the multimodal layer with two additional unimodal areas with 30 units each (*13620* trainable parameters).

**Training.** Prior to training we initialised each layers weights uniformly between $-k$ and $k$ where:

$$k = \sqrt{\frac{1}{N_{\text{inputs}}}} \tag{16}$$

To calculate the loss per batch, we summed each readout unit's activity over time, per trial, then applied the log softmax and negative log likelihood loss functions. Network weights were trained using Adam [27] with the default parameter settings in PyTorch: lr = 0.001, betas = (0.9, 0.999), and no weight decay. In the backward pass we approximated the derivative using the SuperSpike surrogate function [16] with a slope $\sigma = 10$.

## Analysis

**Accuracy filter.** In Results: Nonlinear fusion excels in naturalistic settings, we randomly sampled detection task parameters and evaluated the accuracy of the linear and nonlinear fusion algorithms. To filter out trivial or excessively challenging trials, we used the performance of a majority class classifier (i.e. 0.7 accuracy if 70% of trials share a single label) to define an accuracy filter. Here, $a$ represents the accuracy of this classifier. We define $w = 1 - a$ as a measure of the remaining variability unexplained by the classifier, indicating the potential for algorithm improvement over this strict baseline. We retained results only where both of the following conditions hold:

- max(linear accuracy, nonlinear accuracy) $> a + w/8$

- min(linear accuracy, nonlinear accuracy) $< 1 - w/8$

The first condition ensures that the best algorithm is performing reasonably well. The second ensures that the performance of the two algorithms is not saturated. This ensures that both algorithms are evaluated under conditions that are neither overly trivial nor challenging, allowing a more meaningful comparison of their performance.

**Random forest regression.** In Results: Nonlinear fusion excels in naturalistic settings, we show the detection tasks parameters *importances* in predicting the difference in accuracy between **linear** and **nonlinear** fusion. To compute these, we trained a random forest regression algorithm to predict the accuracy difference from the task parameters. Then use (impurity-based) feature importances to estimate how informative each parameter is in predicting this difference [28].

**Shapley values.** In Results: Unit ablations demonstrate a causal link between additivity and network behaviour, we used Shapley Value Analysis to measure the causal roles of individual spiking units in multimodal networks trained on different tasks. The method was implemented in [21], derived from the original work of [29]. Shapley values are a rigorous way of attributing contributions of cooperating players to a game. Taking into account every possible *coalition*, we can determine the precise contribution of every player to the overall game performance. This however becomes quickly infeasible, as it scales exponentially with the number of

elements in the system. We can however estimate it by sampling random coalitions, to then approximate each element's contributions (Shapley values). In our case we consider individual neurons (players) performing task inference (the game) following a *lesion* (where the coalition consists of the *un-lesioned* neurons).

## Supporting information

**S1 File. Contains: Additional text, figures A-F, sample Python code for each task and our mathematical derivations.**
(PDF)

## Acknowledgments

We thank Curvenote for their support in formatting the manuscript, Nicolas Perez-Nieves for his help in writing the initial SNN code, and members of both Laboratoire Jean Perrin and the Neural Reckoning lab for their input.

## Author Contributions

**Conceptualization:** Marcus Ghosh, Gabriel Béna, Volker Bormuth, Dan F. M. Goodman.

**Data curation:** Marcus Ghosh, Gabriel Béna, Dan F. M. Goodman.

**Formal analysis:** Marcus Ghosh, Gabriel Béna, Dan F. M. Goodman.

**Funding acquisition:** Marcus Ghosh, Volker Bormuth, Dan F. M. Goodman.

**Investigation:** Marcus Ghosh, Gabriel Béna, Volker Bormuth, Dan F. M. Goodman.

**Methodology:** Marcus Ghosh, Gabriel Béna, Volker Bormuth, Dan F. M. Goodman.

**Project administration:** Marcus Ghosh, Gabriel Béna, Volker Bormuth, Dan F. M. Goodman.

**Resources:** Marcus Ghosh, Gabriel Béna, Volker Bormuth, Dan F. M. Goodman.

**Software:** Marcus Ghosh, Gabriel Béna, Dan F. M. Goodman.

**Supervision:** Volker Bormuth, Dan F. M. Goodman.

**Validation:** Marcus Ghosh, Gabriel Béna, Dan F. M. Goodman.

**Visualization:** Marcus Ghosh, Gabriel Béna, Dan F. M. Goodman.

**Writing – original draft:** Marcus Ghosh, Gabriel Béna, Volker Bormuth, Dan F. M. Goodman.

**Writing – review & editing:** Marcus Ghosh, Gabriel Béna, Volker Bormuth, Dan F. M. Goodman.

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
