## [Decision Letter · Decision Letter 0]

22 Feb 2024

Dear Dr Ghosh,

Thank you very much for submitting your manuscript "Multimodal units fuse-then-accumulate evidence across channels" for consideration at PLOS Computational Biology.

As with all papers reviewed by the journal, your manuscript was reviewed by members of the editorial board and by several independent reviewers. In light of the reviews (below this email), we would like to invite the resubmission of a significantly-revised version that takes into account the reviewers' comments.

We cannot make any decision about publication until we have seen the revised manuscript and your response to the reviewers' comments. Your revised manuscript is also likely to be sent to reviewers for further evaluation.

Sincerely,

Tianming Yang

Academic Editor

PLOS Computational Biology

Thomas Serre

Section Editor

PLOS Computational Biology

Reviewer's Responses to Questions

**Comments to the Authors:**

Reviewer #1: In the present study, the authors investigated two conceptual frameworks for multisensory decision making, fuse-then-accumulate (FtA) and accumulate-then-fuse (AtF), on a series of novel multimodal tasks. By design, these “comodulation” tasks entail coincidence detection between channels, and thus require the “multimodal” unit in FtA, as shown by an excelled performance of trained spiking neural networks with FtA on the comodulation tasks, but not on the classical tasks. Combining simulation with an ideal observer model, the authors provided a deeper understanding of why the nonlinearity embedded in the FtA framework is important and how the signal sparsity plays a role. Finally, the authors explored the link between single neuron properties and the super-additive nonlinearity in FtA. Together, the authors argue that the FtA framework is optimal for a wide range of multisensory problems and may be widely adopted by biological systems.

The manuscript is overall well-organized, the analyses are thoughtful, and the methodology is sound. The discussions of the series of comodulation tasks are also insightful. However, I have several major issues, mainly related to interpreting the results and placing the claims properly in the context of previous literature, along with other minor comments.

Major:

1. FtA is not new.

i) In fact, the large body of early multisensory studies, such as those from Barry Stein and colleagues, all point to the FtA framework. For instance, the superadditivity and inverse effectiveness of multimodal units in SC have been suggested to facilitate near-threshold (i.e., ”sparse”) signal detection (see, e.g., Stein and Stanford, 2008, https://doi.org/10.1038/nrn2331), very similar to what the authors propose here.

ii) Even Fetsch et al, 2013 and the series of work from Angelaki and DeAngelis did not suggest an AtF algorithm (Line 46). Although they did not report a superadditivity in cortical areas, their working hypothesis was still FtA, in the sense that visual and vestibular self-motion cues are first fused in multisensory areas such as MST and then accumulated over time in decision making areas such as LIP. (But see a follow-up study suggesting “accumulate-while-fuse” or “late-convergence” in the same system: Hou et al., 2019, https://doi.org/10.1016/j.neuron.2019.08.038)

iii) Similarly, I don’t think Coen et al., 2023 has suggested AtF either (Line 50). It is clearly proposed in their Figure 7 that MOs first fuses visual and auditory cues, and then a downstream integrator accumulates MOs activity, a framework consistent with FtA.

iv) The debate between FtA and AtF is also not new, which should not be overlooked in this study. See for example Bizley et al., 2016, https://doi.org/10.1016/j.conb.2016.06.003.

Together, it is inappropriate to claim that the authors “propose a new FtA algorithm” (line 65). Rather, the present study provides, at most, detailed analyses of tasks where coincidence detection of multimodal signals is important and where fusion before accumulation is required. The novelty is not the FtA framework per se, but the generalization of “coincidence detection” from near-threshold stimuli in terms of signal amplitude to those in terms of temporal sparsity. The manuscript, especially the Abstract, Introduction, and Discussion, should be rewritten accordingly.

2. Taking a step back, the dichotomy between FtA and AtF may itself be debatable, especially in the context of distributed computation across interconnected brain regions. From early sensory processing to perceptual decision making, multiple processes of “fusion” and “accumulation” could occur at different hierarchies and on different timescales. In this case, how can one define the order of their occurrence? In addition, as suggested in Hou et al., 2019 and Coen et al., 2023, they could also happen simultaneously in the same neural circuit. The authors should therefore discuss the limitations of the oversimplified network architectures in Figure 1, and provide a stronger link to biological reality by maybe suggesting where experimentalists should test the predictions of the present study in the brain.

3. The connection between Section 3.5.2 and the rest of the paper is not very clear. How does the inverse effectiveness of single cells contribute to the nonlinearity needed to solve the task? What is the relationship between single cell nonlinearity and network nonlinearity? Why do we need both? In particular, in Figure 4C, why is the superadditivity of the Cl and Dd tasks even larger than that of the Cm+ task? In Section 3.5.2 and Figure 4B, single neurons with shorter tau were associated with higher additivity values, whereas in Figure 8.S6, additivity is positively correlated with tau in the Cm+ task. Why is this?

Minor:

1. Figure 1.C2: in the probabilistic version of the comodulation task, what is the correct answer for a trial where L > R but LL < RR?

2. Line 132: what do you mean by ceiling accuracy? How could the performance better than it?

3. Line 144: how does “sparsity of the signal” depend on p_e and p_c? Is it “sparsity of the correct signal”?

4. Equation (1): please provide some intuition as to why nonlinearity is important for solving a sparse, coincidence-detector-like task.

5. Section 3.5.2: method Section 5.4.1 should be mentioned here.

6. Figure 5: each trial’s alpha (transparency?) is not distinguishable.

Reviewer #2: Overall I think this manuscript makes a valuable contribution to the multisensory research field and will be of wider interest to sensory scientists interested in how cues are fused. The papers strength is in tackling a problem at 4 different levels, each of which supports and extends the conclusions of the other and at the end generates testable biological hypothesis. By implementing minimal artificial neural networks, Bayesian ideal observer models, spiking neurons and spiking neural networks on an identical family of “multisensory” tasks the authors are able to demonstrate a specific set of stimulus conditions in which it is advantageous to fuse information across “senses” before accumulating information. By exploring networks of spiking neurons and ablating targeted populations of cells they are able to demonstrate that in stimulus conditions in which fusing before accumulating (FtA) was advantageous model performance was dependent on neurons that integrated information over short timescales (i.e. coincidence detectors).

While the work is certainly interesting, and appears performed to a high standard, there are a number of places where some additional clarification would be helpful (especially for biologists; the paper is very much written for computational experts at the moment).

Generally, I found the manuscript to be posed at quite a high level; probably a computational expert it is clear, but as a biologist I found several places (e.g. section 3.4) where some additional explanations would have been really helpful. A paragraph at the beginning of the results that provided a big picture roadmap of the approach to be taken and what would be learned by each would provide a framework in which to hang the subsequent results. I also found it quite confusing to go from networks, to algorithms and then back to networks; terminology is not always clear. For example, how are the ‘multimodal’ model architectures are different from the ‘fuse then accumulate’ – they seem like the same model from figure 1 and the text in sections 3.1 and 3.2 doesn’t differentiate them, but I’m pretty sure the later is an ideal observer model (as in section 3.3) and the former an artificial neural network? Or do these not relate to the original networks developed for Figure 1?

Figure 1C & 2F,G – what is time here? The time to a decision? The timebin over which integration is performed? What are the units?

In the comodulation task it seems obvious that to solve the task there must be some element that looks across both channels with sufficient temporal resolution to pick up coherent signals; by definition (unless accumulation is very short-term) this is not going to be possible with a AtF model (as evidenced by chance level performance in the comodulation task (Fig.1C2)). Its not clear to me how this problem is overcome by linear integration when the signals are denser; can the authors provide some intuition as to what it is in the signal domain that changes? Or is the direction not signalled via comodulation here? I think the lines “E” in whether the signal is emitting or not but its not obvious how an observer knows what is signal vs noise (unless its via comodulation, but then why does E=1 cover the comodulated and unisensory events?). Is it simply that due to the probabilistic nature of the stimulus generation there genuinely is slightly more signal in the comodulated direction than the null direction and that making the signal denser highlights this difference? Or am I missing something fundamental…?

Relatedly, have you explored temporal windows of integration in the earlier models? Fusing and then accumulating allows a network to work on both short and longer timescales (as the network analysis later in the paper points out, and which is required if you want to both remember the past and detect coincident events across channels). Presumably at some point if evidence accumulation is sufficiently short-term it effectively becomes coincidence detection and therefore both the timescale over which the stimuli unfold and the timescale over which an evidence accumulation centre accumulates unisensory information together dictate to what extent there is a benefit to a prior fusion stage? Is there any way of exploring this parameter to relate it to the time constants within the spiking neural networks?

In figure 5 it seems like the really interesting trials are those highlighted by the arrows in the dense detection task. What is it about those trials that result in opposing predictions; are they trials with particularly high/low numbers of comodulated signal intervals or anything interpretable like that? Could a psychophysicists probe these trials specifically to understand the strategies of their observers?

There are some really trivial changes that would ease the cognitive burden when reading the manuscript – for example on figure 5, why not label the columns Classic / Sparse / Dense / Comodulated? On the delta accuracy plots, specify that this is a difference in % correct etc.

The discussion could make it clearer that these results relates to a very specific subset of multisensory decision making tasks that require evidence accumulation rather than other temporally restricted, but common, multisensory tasks (such as sensory illusions, lip reading etc).

While I agree that most experimental studies of evidence accumulation have focussed in the idea or indeed found evidence in favour of accumulating within a modality and then fusing across modalities, it (1) feels a little inaccurate to say this is a canonical perspective (line 44) for all multisensory tasks when there is now very good evidence that there aren’t really ‘unisensory’ areas in cortex and (2) it feels a little disingenuous to try and claim that this is the first suggestion of a FtA perspective (line 65). This is certainly the first rigorous computational exploration of the two possibilities, but I think its far from the first study to suggest multisensory information might be integrated prior to a decision-making step.

Line 101 – I think this should be trial’s rather than trials?

Reviewer #3: Ghose et al. present an interesting ML-flavored treatment of multisensory integration and coincidence detection in spiking networks. The approach is fairly novel and should be quite useful for understanding certain kinds of information integration in the nervous system. I especially like the connection to prey capture, and the generalization to cases where Nc channels are used to discriminate Nd classes.

I’m less convinced (1) that they have provided the best/only explanation for *why* multimodal units exist in the brain (though the question is an important one, given the logic they lay out about linearity and the decision stage). I also (2) disagree that the ‘classical’ paradigm fails to apply outside of the laboratory, or that comodulation/coincidence detection is more naturalistic; indeed it may be implausible without some modifications to account for the temporal mismatch across neural representations for most modality pairings. Lastly, (3) I think their exposition at the outset —particularly the dichotomies of AtF vs. FtA and classical vs. comodulated — is incomplete at best, and may be perplexing or alienating to a significant portion of the field. I’ll explain what I mean below. Otherwise, the paper is well written and technically impressive, with elegant analyses and nice figures, so I’m rather positive about the contribution.

I’ll tackle the above three general comments in reverse order:

(3) First I think the setup and assumptions here may reflect a misreading of the literature. Arguably the ‘classic’ approach dates back to Landy, Maloney, Jacobs and others in the 90’s, with

---

## [Decision Letter · Decision Letter 1]

11 Jun 2024

Dear Dr Ghosh,

We are pleased to inform you that your manuscript 'Nonlinear fusion is optimal for a wide class of multisensory tasks' has been provisionally accepted for publication in PLOS Computational Biology.

Best regards,

Tianming Yang

Academic Editor

PLOS Computational Biology

Thomas Serre

Section Editor

PLOS Computational Biology

Reviewer's Responses to Questions

**Comments to the Authors:**

Reviewer #1: The authors have responded thoughtfully to all my concerns, and the manuscript has been revised to more accurately reflect the contributions of this work. I only have one more minor comment.

The first sentence of the Discussion, “Prior experimental and theoretical work suggests that multimodal neurons linearly fuse unimodal information across channels”, is still overly general and a bit misleading, although the “super-additive” neurons from Stanford and Stein’s work are now properly mentioned in the Introduction of the revised manuscript. I would suggest that the authors revise this sentence, and, even better, try to discuss how the behavioral tasks considered in the present paper relate to the computations that the super-additive neurons could solve in the classical multisensory literature. For instance, in early sensory areas such as SC, super-additivity was thought to be relevant for detecting multisensory stimuli that are near-threshold (in terms of signal amplitude) but occur coincidentally. The present study somewhat echoes this early interpretation, with the key difference (or generalization) being the meaning of “near-threshold” — from the threshold of amplitude to that of the temporal sparsity.

Reviewer #2: I found the revised manuscript to be much clearer in its writing and motivation, and the subsitution of the terms 'nonlinear' and 'linear' fusion (rather than fuse then accumilate vs. acculmilate then fuse) to be a great improvement. The authors have addressed all of my concerns satisfactorily and I congratulate them on a great study!

Reviewer #3: I thank the authors for the thoughtful responses. The reframing makes sense and I have no further comments on the revised manuscript.

**Have the authors made all data and (if applicable) computational code underlying the findings in their manuscript fully available?**

Reviewer #1: Yes

Reviewer #2: Yes

Reviewer #3: Yes

PLOS authors have the option to publish the peer review history of their article (what does this mean?). If published, this will include your full peer review and any attached files.

Reviewer #1: **Yes: **Han Hou

Reviewer #2: No

Reviewer #3: **Yes: **Christopher Fetsch

---

## [Editor Report · Acceptance letter]

28 Jun 2024

PCOMPBIOL-D-24-00023R1 

Nonlinear fusion is optimal for a wide class of multisensory tasks

Dear Dr Ghosh,

I am pleased to inform you that your manuscript has been formally accepted for publication in PLOS Computational Biology. Your manuscript is now with our production department and you will be notified of the publication date in due course.

With kind regards,

Olena Szabo
